# Structure and Junctional Complexes of Endothelial, Epithelial and Glial Brain Barriers

**DOI:** 10.3390/ijms20215372

**Published:** 2019-10-29

**Authors:** Mariana Castro Dias, Josephine A. Mapunda, Mykhailo Vladymyrov, Britta Engelhardt

**Affiliations:** Theodor Kocher Institute, University of Bern, 3012 Bern, Switzerland; josephine.mapunda@tki.unibe.ch (J.A.M.); mykhailo.vladymyrov@tki.unibe.ch (M.V.)

**Keywords:** brain barriers, blood-brain barrier, neurovascular unit, blood-cerebrospinal fluid barrier, arachnoid barrier, glia limitans, tight junctions, adherens junctions

## Abstract

The homeostasis of the central nervous system (CNS) is ensured by the endothelial, epithelial, mesothelial and glial brain barriers, which strictly control the passage of molecules, solutes and immune cells. While the endothelial blood-brain barrier (BBB) and the epithelial blood-cerebrospinal fluid barrier (BCSFB) have been extensively investigated, less is known about the epithelial and mesothelial arachnoid barrier and the glia limitans. Here, we summarize current knowledge of the cellular composition of the brain barriers with a specific focus on describing the molecular constituents of their junctional complexes. We propose that the brain barriers maintain CNS immune privilege by dividing the CNS into compartments that differ with regard to their role in immune surveillance of the CNS. We close by providing a brief overview on experimental tools allowing for reliable in vivo visualization of the brain barriers and their junctional complexes and thus the respective CNS compartments.

## 1. Introduction

The brain barriers established by the endothelial blood-brain barrier (BBB), the epithelial blood-cerebrospinal fluid barrier (BCSFB), the meningeal brain barriers and the blood spinal cord barrier are essential for maintaining central nervous system (CNS) homeostasis [1]. While the structural and junctional components of the BBB and of the BCSFB of the choroid plexus (ChP) have been vastly explored and described, much less is known about the cells and junctional complexes establishing the meningeal barriers and the glia limitans.

Development of the CNS vasculature begins with the process of vasculogenesis, where angioblasts, which originate from the mesoderm, move to the head region and form the primary perineural vascular plexus around the developing brain. The proliferating neuroectodermal tissue is invaded by vascular sprouts from the primary perineural plexus establishing the brain vasculature by a process called angiogenesis. The unique barrier properties of the brain endothelial cells are not intrinsic and are rather induced by a process referred to as barriergenesis by the continuous crosstalk with the developing neuroectodermal tissue [2,3,4,5].

The ChP develops from different locations along the dorsal axis of the neural tube, with the hindbrain ChP of the fourth ventricle being the first to be formed, once the neural tube is closed [6]. While the ChP stroma develops from mesenchymal cells, the ChP epithelium derives from neuroepithelial cells. Despite the different origin along the rostral-caudal axis of the developing nervous system, the mature ChP epithelia are morphologically similar (reviewed in Reference [6]).

The meningeal barriers comprise three layers: the dura mater, the arachnoid mater and the pia mater, and cover the brain and the spinal cord from its earliest stages of development, establishing a protective covering of the CNS in the adult. Our current understanding of the development of the meninges has not much advanced beyond identifying the origins of pial, arachnoid and dural fibroblasts in the frontal brain from neural crest, and in the midbrain, hindbrain and spinal cord from mesoderm [7]. Despite their differences in developmental origin, the cells of the pia, arachnoid and dura mater have been described to establish functionally similar barriers. However, the respective barrier function may be established by different molecular components of their junctional complexes [8,9,10,11].

Taken together, junctional complexes established between cells of the brain barriers might not only differ between the different barriers but also within one respective barrier along the rostral-caudal axis. Here, we summarize the current knowledge of the brain barriers, with a particular focus on the molecular components of their junctional complexes. Additionally, we suggest that visualization of brain barriers junctional complexes provides useful landmarks for in vivo imaging of immune surveillance of the CNS.

## 2. Blood-Brain Barrier and Its Central Players: Cellular and Acellular Components of the Neurovascular Unit (NVU)

The brain vasculature is a big and complex network composed of arteries and arterioles, capillaries, venules and veins, which allows for the vital distribution of nutrients and oxygen to the CNS at the level of the microvasculature, namely arterioles, capillaries and post-capillary venules [12]. Brain microvascular endothelial cells, referred to as the blood-brain barrier (BBB), possess unique features restricting the free passage of ions, molecules and cells from the blood into the CNS parenchyma, while facilitating the transport of toxic compounds out of the CNS [13,14]. Development and maintenance of BBB characteristics in CNS microvascular endothelium relies on the continuous crosstalk between cellular and acellular elements of the CNS. In the adult, the entity of BBB endothelium, pericytes, astrocytes, and the basal membranes are thus collectively referred to as the neurovascular unit (NVU) (represented in Figure 1) [15]. Maintenance of a functional NVU is a prerequisite for BBB function and requires understanding of the specific contributions of these cells to barrier function.

### 2.1. Endothelial Cells

Most studies of the BBB focus on the capillary endothelial cells, due to their unique properties [16,17]. Besides lacking fenestrae, these endothelial cells are brought together by the presence of continuous and complex tight junctions that prohibit free diffusion of molecules across the BBB via the paracellular route [18,19,20]. Additionally, they have very low pinocytosis rates when compared to the peripheral endothelium [21,22]. Of notice, the major facilitator superfamily domain-containing protein 2 (Mfsd2a), a membrane transport protein, is specifically expressed in the CNS endothelium and acts as a key regulator suppressing vesicular activity at the BBB [23,24]. To ensure the passage of nutrients from the blood into the CNS and to remove potential toxic agents from the brain, the BBB endothelial cells express specific transporters and efflux pumps [25,26,27]. In particular, the efflux pump MDR1/P-glycoprotein (Pgp) translocates potentially harmful lipophilic or endogenous molecules from the CNS to the blood [28]. The glucose transporter GLUT-1 (*SLC2A1)* is also highly enriched in the BBB endothelium, allowing for glucose delivery to the CNS [29]. The barrier characteristics referred to above are also present at the level of the endothelial cells forming the BBB in the post-capillary venules. While the capillaries represent a barrier for solutes and ions, leukocyte trafficking is regulated at the level of the post-capillary venules where the endothelial cells express specific adhesion molecules [1,30].

Additionally, the luminal side of the brain endothelium is covered by the glycocalyx, a carbohydrate-rich mesh of anionic polymers, which acts as a first physical barrier between the blood and the vessel wall [31]. In peripheral microvessels, the glycocalyx regulates vascular permeability to water and to albumin, and glycocalyx breakdown results in plasma leakage and enhanced leukocyte recruitment, in vivo [32,33]. Recent advances in in vivo imaging have allowed to confirm barrier forming role of the glycocalyx in peripheral vessels [34] and at the level of the BBB [35].

However, there are areas of the brain, the circumventricular organs, where the capillaries lack BBB properties and rather have fenestrae allowing blood components to freely pass towards specialized neurons serving neuroendocrine and neurosensory functions [36]. In addition, the ChP vasculature is characterized by fenestrae allowing free access of blood components into the ChP stroma [37].

### 2.2. Mural Cells: Smooth Muscle Cells and Pericytes

Mural cells comprise smooth muscle cells and pericytes. While pericytes, embedded in the endothelial basement membrane, incompletely cover the capillaries, smooth muscle cells seem to completely surround arterioles and to a lower degree venules and, together with the endothelial basement membrane, compose the tunica media [38]. Additionally, smooth muscle cells express contractile proteins such as alpha-smooth muscle actin, which allows for the control of vessel diameter and blood flow [38,39,40,41]. Recent studies have assigned additional roles of pericytes in regulating blood flow at the level of capillaries [42], which has initiated a debate on the role of arteriolar smooth muscle cell or capillary pericyte contractility in regulating regional cerebral blood flow. At the same time, pericytes are morphologically and functionally distinct from smooth muscle cells [43], which is further supported by the recent discoveries on their different gene expression profile in CNS blood vessels [44]. Pericytes interact with the endothelial cells via specific adhesion points that can either form peg-and-socket junctions, due to the presence of N-cadherin [45], or can occur as adhesion plaques, gap junctions and even tight junctions [46,47,48]. In the brain, endothelial coverage by pericytes is extremely high, with an endothelial cell/pericyte ratio between 1:1 to 3:1 [49]. Additionally, pericytes are involved in wound healing, angiogenesis, deposition of extracellular matrix and, during embryogenesis, have a strong role in BBB formation by suppressing vesicular activity in brain endothelial cells [43,50,51]. In the adult stage, loss of pericytes impairs BBB properties of endothelial cells by affecting their gene expression profile leading to increased vesicular trafficking and loss of polarization of astrocytic endfeet [51]. Moreover, pericytes regulate expression of cellular adhesion molecules in the BBB endothelium [50,51,52].

### 2.3. Basement Membrane

Pericytes and endothelial cells secrete and are surrounded by the extracellular matrix proteins that comprise the endothelial basement membrane. It is a crosslinked network composed of type IV collagen, α4 and α5 laminins, nidogen, heparan sulphate proteoglycans and some glycoproteins [53]. A unique feature of CNS microvessels is the presence of a second basement membrane, referred to as parenchymal basement membrane, which is secreted by astrocytes [53,54]. With a composition of α1 and α2 laminins at the brain surface and with α2 laminins at the perivascular spaces, the parenchymal basement membranes are molecularly distinct from the endothelial basement membrane [55]. At the capillary level it merges with the endothelial basement membrane, while at the level of post-capillary venules there is a small perivascular space between those two basement membranes. The three-dimensional configuration and the components of these basement membranes host specific signaling cues for cellular growth and survival during the development and further maintenance of the BBB, while giving physical support to the vasculature [56] (reviewed in Reference [57]).

### 2.4. Astrocytes

Astrocytes also assume a tight relationship with the endothelium and the mural cells. These glial cells are widely present throughout the CNS and perform a supportive role that regulates the neuronal microenvironment by controlling neurotransmitters and electrolytes balance, synapse formation and clearance of axonal debris [16,58,59]. Furthermore, they have extensive cellular protrusions that cover blood vessels and neuronal synapses, promoting a direct link between neuronal circuits and the vasculature [16]. The astrocytic endfeet are in direct contact with the microvasculature of the CNS, covering a large part of the abluminal aspect of CNS microvessels [39]. Astrocytic processes express a range of proteins, including α-dystroglycan and dystrophin, which link the astrocyte endfoot to the parenchymal basement membrane, and the water channel aquaporin 4, which plays a crucial role regulating water homeostasis in the CNS [60,61]. Astrocytes have been described as an important intermediate in promoting endothelial cell barrier properties, as well as secreting the components of the parenchymal basement membrane, as mentioned above [16,53,62]. Additionally, the astrocytic endfeet and the parenchymal basement membrane form a second barrier of the CNS called glia limitans [1]. Several in vitro studies showed that with a co-culture of astrocytes and endothelial cells, the junctional complexes between adjacent endothelial cells upregulate its main proteins, contributing to an increase in barrier integrity [63,64]. Moreover, secretion of sonic hedgehog, retinoic acid and angiopoietin-1 by astrocytes supports the barrier properties of brain endothelial cells [65,66]. On the other hand, the interaction of astrocytes with endothelial cells is also very important, with endothelial-secreted factors such as LIF1 promoting astrocytic differentiation [67].

## 3. Molecular Constituents of the BBB

The BBB acts as a physical and metabolic barrier due to the unique biochemical make-up of the BBB endothelial cells. Here we will only focus on describing the molecules establishing the paracellular diffusion barrier of the BBB, which are proteins composing tight junctions, adherens junctions, transmembrane proteins localized at cell-to-cell contacts outside the organized tight junctions and adherens junctions, and intracellular junctional scaffolding proteins.

### 3.1. BBB Tight Junctions (TJs)

TJs localized between adjacent endothelial cells are core elements actively involved in the establishment of a paracellular barrier, which limits free diffusion of ions and molecules at cell-cell junctions, adopting a “gate” function. TJs were discovered by transmission electron microscopy studies and described as focal points were the membranes of adjacent cells come together, obliterating the intercellular cleft [68]. Freeze-fracture electron microscopy showed that the TJ strands of the BBB endothelium resemble rather those of epithelial cells by producing predominantly protoplasmic (P-face) associated continuous and complex particle strands [69,70,71,72,73,74]. Although the exact mechanism underlying P-face and exoplasmic (E-face) association of TJ particles remain to be shown, occludin and claudins are localized to the TJ particle strands and thus it is hypothesized that a high fraction of P-face associated TJs correlates with increased avidity of the transmembrane TJ proteins to the actin cytoskeleton elements [73].

Besides blocking paracellular diffusion, TJs are proposed to act as a “fence” by creating an intramembrane diffusion barrier. This not only stops free passage of proteins and lipids across the lipid bilayer [75], but also prohibits an intermixing of the components of the plasma membrane, creating distinct apical and basolateral membrane sites [76]. Additionally, they distribute selected membrane components to the cell surface and allow for accumulation of internal scaffolding proteins, essential to establish a link to the cytoskeleton [77]. Altogether, the “gate and fence” function that TJs assume is essential to assure the regulation of a physiological barrier. Most of the gathered knowledge about TJs is derived from epithelial cells and readily assumed to hold true for endothelial cells, despite lacking formal proof [78]. In terms of molecular composition, TJs are composed of transmembrane proteins, cytoplasmic plaque proteins, signaling proteins and adapters that bind these complexes to the actin cytoskeleton. Transmembrane proteins are suggested to be the elements playing the barrier and fence role, since they are constituted by transmembrane, cytoplasmic and extracellular domains. Three groups of transmembrane proteins are found to be localized in TJs: the claudins, the immunoglobulin superfamily junctional adhesion molecules (JAMs) and the tight junction-associated MARVEL proteins (TAMP) [78,79].

Claudins are the major components of TJs and to date, 27 members of this family have been identified in mammals [80]. Their general structure consists of an intracellular NH_2_ domain, four transmembrane domains, two different extracellular loops (ECL1 and ECL2) and one intracellular COOH terminus [79]. In their C-terminus, most of the claudins possess a PDZ-binding motif used to interact with the TJ-associated scaffolding proteins [81]. ECL1 is involved in regulating the paracellular tightness and ion permeability, while ECL2 mediates cis/trans claudin-claudin interactions [82,83]. In vitro data showed that in contrast to other TJ transmembrane proteins, claudins are sufficient to form TJs strands in fibroblasts and that paracellular permeability is increased upon disruption of claudins, which suggests that this family of proteins is essential for barrier formation [84,85,86,87]. It has been suggested that specific barrier functions demand expression of different combinations of claudins, which is believed to influence the charge and size-selective properties of the barrier [88]. Claudins such as claudins-1, -3, -5, -11, -14 and -19 act as TJ sealing membrane proteins, while claudins-2, -10, -15 and -17 are involved in ion pore formation between two adjacent cells [89]. Claudin-deficient mice have been extremely helpful to understand the specific function of individual claudins in specific tissue barriers. For example, claudin-1 deficiency leads to epidermal permeability barrier defects in the skin, while lack of claudin-11 causes defects in CNS myelin sheaths [90,91]. Regarding the BBB endothelium, claudin-1, claudin-3, claudin-5, claudin-11 and claudin-12 were described to be expressed in the BBB endothelial cells [92,93,94,95]. Claudin-5 is specifically expressed in endothelial cells with very high expression in brain endothelium [50]. Lack of claudin-5 leads to perinatal death in mice, due to exacerbated BBB permeability to tracers smaller than 800 Da [92,96]. Supporting the important role of claudin-5 in BBB TJs, inducible knock-down of claudin-5 in adult mice was found to lead to seizures and behavioral changes, due to loss of BBB properties [97]. Despite this dramatic failure of BBB TJ functions, mice lacking claudin-5 still present morphologically intact and P-face associated TJs in the BBB, which suggests that claudins other than claudin-5 maintain BBB TJs in these mice [92]. Claudin-3 has been described as an additional component of BBB TJs, mediating TJ maturation during development by acting as a downstream target of Wnt/β-catenin signaling [50,98,99]. Additional evidence for a role of claudin-3 in regulating BBB integrity was derived from studies in an animal model for multiple sclerosis or in human glioblastoma multiforme, were claudin-3 immunostaining was found to be specifically lost in inflamed CNS microvessels [98]. These observations supported a role for claudin-3 in maintaining junctional BBB integrity. Making use of a novel claudin-3 deficient mouse and advanced transcriptomic profiling of brain endothelial cells we and others recently made the unexpected observation that claudin-3 mRNA cannot be detected in brain endothelial cells and thus claudin-3 is not expressed in brain endothelial cells and does not contribute to BBB TJs [100,101,102]. Disturbingly, we found most of the claudin-3 detecting antibodies to produce comparable junctional immunostainings of brain endothelial cells in wild-type and claudin-3-deficient mice suggesting that available claudin-3 antibodies cross-react with another junctional molecule and do not allow to the reliable detection of expression and junctional localization of claudin-3 protein [102]. In addition, positive immunostaining for claudin-1 and claudin-11 has been reported in mouse brain endothelium [103] [93,95]. However, we and others showed that claudin-1 mRNA is not detectable in brain endothelial cells [44,104,105]. In fact, endothelial specific inducible ectopic expression of claudin-1 in BBB endothelium reduces BBB leakiness in an animal model of multiple sclerosis and ameliorates clinical signs of the disease further supporting absence of claudin-1 at the BBB [104]. Similarly, claudin-11 mRNA was not detected in brain endothelial cells in single-cell RNA sequencing datasets from mouse brain microvessels, but its expression was readily detected in oligodendrocytes [44,105].

Finally, claudin-12 is an atypical member of the claudin family because it does not have a PDZ-binding motif [69]. Prominent junctional immunostaining for claudin-12 [92] suggested its localization to BBB TJs. However, by establishing a claudin-12 deficient mouse we could recently show that expression of claudin-12 in microvascular brain endothelial cells is low to absent and is not required for barrier function of the BBB [106]. Analysis of the localization of claudin-12 protein in brain endothelial cells was again hampered by the lack of antibodies specifically detecting claudin-12 in wild-type but not in claudin-12 deficient mice.

Taken together, reports describing junctional localization of claudin-1, claudin-3, and claudin-12 proteins in brain endothelial cells may be inaccurate due to not well-defined cross-reactivities and thus false-positive staining of claudin antibodies. Cross-reactivity of the anti-claudin antibodies may be due to the high degree of conservation and a common amino acid sequence within some members of the claudin family, which could potentiate a cross-reactivity of anti-claudin antibodies [83]. Furthermore, recent studies described detection of claudin-11 at the protein level in BBB endothelial cells in mice especially in the spinal cord and man [93,94]. As single-cell RNAseq analysis of brain endothelial cells failed to detect claudin-11 in these cells [44,105] and so, it is mandatory to explore if claudin 11 mRNA levels are too low for detection in the single-cell RNAseq screens or if detection of claudin-11 protein in these studies is again due to the cross-reaction of claudin-11 antibodies with other junctional entities.

Thus, at present, claudin-5 is the only claudin confirmed to be expressed at the mRNA and protein level in brain endothelial cells, to be localized to BBB TJs and to play a major role in BBB TJs integrity. Advanced transcriptome profiling studies have identified additional claudins, e.g., claudin-25/CLDND1 to be expressed in brain microvascular endothelial cells, with their exact function to be determined [44,105].

The TAMP family comprises the proteins occludin, tricellulin and MARVELD3 [107,108,109], which have four transmembrane domains and a MARVEL (MAL-related proteins for vesicle trafficking and membrane link) domain. To date, occludin and tricellulin were found to be expressed at the TJs of the BBB [110,111]. Occludin, the first transmembrane protein discovered to be localized exclusively in TJs, is highly expressed in endothelial cells of the CNS and was suggested to play a role in barrier integrity [107,110,111,112,113]. However, occludin-deficient epithelial cells develop morphologically intact TJs and consequently maintain barrier function [114]. On the other hand, phosphorylation of occludin was shown to regulate endothelial permeability and to be necessary for cortical actin organization in various endothelial cell models [115,116,117]. Additionally, occludin-deficient mice do not show impaired TJs, but are characterized by chronic inflammation, poor TJ integrity and by calcifications in the brain [118]. These results suggest that occludin may be involved in calcium movement across the BBB and in the regulation of TJ stability and barrier function, rather than TJ assembly. Another member of the TAMP family is tricellulin, which, as the name indicates, is specifically localized to tricellular junctions, a point in which three adjacent cells meet (reviewed in Reference [119]). Albeit being widely described in epithelial monolayers, recently expression of tricellulin was described in endothelial cells forming the BBB and the blood–retinal barrier [111,120], underscoring the epithelial nature of these endothelial barriers. The precise function of tricellulin in endothelial cells forming the BBB and blood–retinal barrier remains however to be described.

The third group of proteins found in BBB TJs are the members of the JAM family, JAM-A, JAM-B and JAM-C. These proteins are immunoglobulin (Ig)-like molecules composed of a single transmembrane domain, two extracellular Ig domains and a PDZ-binding motif in their C-terminus that allows for an interaction with cytoplasmic proteins linking the JAMs to the actin cytoskeleton [121,122]. JAMs are highly enriched in the TJs of epithelial and endothelial cells [123,124,125,126,127] and are the tight junction-associated transmembrane proteins that regulate cell polarity by interacting with Par3 [128,129]. JAM-A immunostaining was found at the BBB and as JAM-A upon transfection into CHO cells establishes a barrier, JAM-A has been suggested to contribute to BBB integrity [126]. It was additionally described to regulate the migration of monocytes across BBB cell-to-cell contacts [130]. Recently, we have described JAM-B to be localized to BBB TJs but discovered that JAM-B-deficient mice have an intact BBB, which suggests that JAM-B is not required for proper BBB function [131]. JAM-B has been found to bind α4β1-integrin and interestingly mediates α4β1-integrin mediated migration of CD8^+^ T cells into the CNS but is not involved in the α4β1-integrin-mediated migration of CD4^+^ T cells across the BBB [131,132,133,134]. The third member of the family, JAM-C was also found to be expressed in brain endothelial cells. Albeit C57BL/6 mice deficient for JAM-C develop a hydrocephalus, this is not due to impairment of BBB function, suggesting that JAM-C is not required for BBB junctional integrity [125,135].

The claudins, occludin and the JAMs expressed in TJs connect to the cytoskeleton via the interaction with intracellular scaffolding proteins, which form the TJ plaque. The membrane-associated guanylate kinase (MAGUK) proteins represent the major subgroup of scaffolding proteins at the TJs [136,137]. These proteins are structurally similar, since they share one or more PDZ domains, an SH3 domain and a catalytically inactive guanylate kinase (GUK) domain and are overall involved in the establishment of cell-cell adhesion, cell polarity and cell survival [138]. Zona occludens (ZO) proteins were the first proteins of the MAGUK family to be identified, with ZO-1 and ZO-2 localizing to endothelial TJs, which is essential for the formation of TJ strands [139,140]. Lack of ZO-1 and ZO-2 in mice is embryonically lethal [140,141]. ZO-1 can be found in the BBB TJs, interacting with claudins, occludin, JAMs and ZO-2, and promoting a link between these proteins and the actin cytoskeleton, by binding to F-actin [142]. It is also a main regulator of tension at vascular endothelial (VE)-cadherin-based adherens junction complexes, mainly binding to α-catenin and recruiting important mediators for junctional assembly and stability [69,143]. Additionally, cingulin, afadin (AF-6) and 7H6 antigen are also involved in the coupling of the junctional complexes to the cytoskeleton, supporting TJ structure and stability [144,145]. Cingulin connects ZO-2, AF-6 and JAMs to F-actin, while 7H6 acts towards maturation and maintenance of TJs [146,147,148].

### 3.2. Adherens Junctions (AJs) of the BBB

Adherens junctions (Ajs) have a role distinct from TJs. Prior to TJ formation, cell-cell contacts are established by AJs, which is a prerequisite for TJ maturation and maintenance. Thus, AJs are generally required for TJ formation, and a continuous crosstalk between AJs and TJs is necessary for the organization and preservation of the junctional complexes (reviewed in Reference [149]). VE-cadherin is the main protein of the endothelial AJs and is involved in blood vessel assembly and endothelial stabilization and survival [150,151]. Its intracellular domains engage p120 catenin and β-catenin [152,153]. By binding to β-catenin, α-catenin acts as a bridge between the cadherins and the actin cytoskeleton, since it binds to vinculin, α-actinin, ZO-1 or formin-1 [154,155]. Interestingly, VE-cadherin promotes expression of claudin-5 in endothelial cells derived from murine embryonic stem cells [156], supporting the notion that mature AJs are prerequisite for the establishment of TJs. Nectin is an additional component of the AJs of the BBB and connects to the actin cytoskeleton via AF-6. The nectin-AF-6 complex contributes to the formation of AJs and stabilization of TJs [157,158].

Additional molecules are found to be localized at the endothelial cell-cell junctions, outside of the organized AJs and TJs. Platelet endothelial cell adhesion molecule-1 (PECAM-1) is a transmembrane protein that belongs to the Ig superfamily. It is highly concentrated and restricted to the endothelial junctions but is found outside the organized TJs and AJs complexes [159,160]. However, it plays a very important role in angiogenesis with a mechanosensory function and is involved in the regulation of vascular integrity as PECAM-1-deficient mice show impaired BBB junctional integrity [161,162,163]. CD99 is an additional protein localized in the endothelial cell junctional complexes, outside of TJs or AJs [164,165]. This protein is a highly O-glycosylated type I transmembrane protein not belonging to any protein family. CD99 mediates leukocyte trafficking across the BBB, with any additional role in BBB junctional integrity remaining to be shown [166]. A schematic representation of the molecular composition of the junctional complexes of the BBB endothelium as known today is illustrated in Figure 2.

Although a large number of molecules localized at the cell-to-cell contacts of the BBB within TJ, AJs and beyond has thus been described, their dynamic interplay and regulation during CNS homeostasis and how their expression and localization changes during neurological disorders when BBB function is impaired remains to be investigated in more detail. In addition, species differences in the expression of molecular constituents of the BBB junctional complexes have been described [94,167,168], an aspect that is important when aiming to translate observations from animal models to the human BBB.

## 4. The Epithelial Blood-Cerebrospinal Fluid Barrier of the Choroid Plexus

The development and maintenance of the CNS is assured by the cerebrospinal fluid (CSF), found in the spinal and brain subarachnoid space, cisterns, sulci and in the cerebral ventricles [169,170]. Importantly, the CSF is produced by the ChP, a highly vascularized secretory tissue that extends into all four cerebral ventricles. The ChP folds out from the ventricular walls and is composed of a highly vascularized stroma that is surrounded by a monolayer of epithelial cells [171]. Because the dense network of capillaries that irrigate the ChP is fenestrated and thus permissive to the passage of blood derived molecules, the epithelial cells surrounding the ChP stroma build a BCSFB [18,172]. Accordingly, the molecules that diffuse from the blood across the ChP vascular wall reach the ChP stroma but are hindered by the epithelial BCSFB to reach the CSF [37].

In accordance to the endothelial BBB, TJs and AJs are present in the BCSFB epithelium, ensuring not only the barrier function but also the apicobasal polarity of the BCSFB [173]. Claudin-1, claudin-2, claudin-3 and claudin-11, together with occludin and ZO-1, were identified to be expressed in the TJs of the BCSFB [100,174,175,176]. JAM-A, but not JAM-B, is found in the epithelium of the BCSFB, while JAM-C is only present in the BCSFB of the ChP of the fourth ventricle [125,131,149]. In the epithelial monolayer, TJs are apically localized and distinguishable from the AJs, which are in the basolateral compartment [149]. Claudin-1-deficient mice die post-natal due to failure of epidermal barrier function [90], suggesting that lack of claudin-1 at the BCSFB is not essential for BCSFB maturation during embryogenesis. However, the precise role of claudin-1 at the BCSFB needs further investigations. Claudin-2 deficient mice show reduced reabsorption of Na^+^ in the proximal tubule of the kidney suggesting a role for claudin-2 in paracellular transport of small cations across the BCSFB [177]. Interestingly, claudin-11 is suggested to induce parallel running TJs strands observed at the BCSFB, but also in myelin sheaths of oligodendrocytes and Sertoli cells, which all express claudin-11 [37,91]. According to the claudin-11 expression pattern, claudin-11 deficient mice display defects in spermatogenesis [91] and show behavioral defects [178]. If any of the CNS related defects is due to lack of claudin-11 at the BCSFB has not been addressed to date [90,91,177,179]. In addition, claudin-3 may be involved in maintenance of BCSFB TJ integrity, since its absence was found to impair BCSFB integrity in a mouse model for multiple sclerosis [180] which was however not confirmed in a second study [102]. Thus, similar to the TJs complexes of the BBB, the precise understanding of the interplay between TJ sealing claudins as claudin-1, claudin-3 and those allowing for transport of water and cations, as claudin-2 in BCSFB TJs, remain to be explored.

Like any other epithelial barrier, the BCSFB epithelial cells establish AJs, where the transmembrane E-cadherin binds to the catenin complex, which anchors the AJs to the epithelial actin cytoskeleton allowing to control adhesive interactions between the BCSFB epithelial cells [149,181].

A schematic representation of the junctional complexes at the ChP BCSFB can be found in Figure 3. In order to set up concentration gradients and assure proper brain function, cellular transporters are also present in the BCSFB. While efflux transporters from the ABC family oversee the return of lipophilic solutes to the blood, solute carrier (SLC) transporters are responsible for the delivery of ions and amino acids to the CSF. Additionally, water flow from the blood to the CSF is in addition to claudin-2 mediated by aquaporin 1 (AQP1) [182].

## 5. The Meningeal Brain Barriers

The meninges constitute three layers, the dura mater, the arachnoid mater and pia mater, which sheath the brain and spinal cord (represented in Figure 4) [183,184]. For decades, the meninges were known for their role in protecting the brain and spinal cord. In recent years, studies have shown that the meninges also play a vital role in the development of the skull and the brain [9,185,186]. Additionally, studies in rats have shown the presence of stomata on the leptomeningeal coverings of the blood vessels in the sub arachnoid space suggesting that the meningeal layers might be a potential route for the transportation of humoral immune factors from the CSF to perivascular or vascular mural spaces [187].

In humans the dura mater is characterized by three different layers, specifically the periosteal, the meningeal and the border cell layer, with the latter forming the border to the arachnoid barrier cells. The dural border cell layer is composed of flattened fibroblasts with scattered intercellular junctions and pronounced extracellular spaces lacking extracellular collagen [10,183,184,188]. It has been shown that the dura and dural border cell layer are immunoreactive to AQP1 [8]. Although junctions have been described between dura mater fibroblasts, their molecular composition is unknown. The dura mater lacks a BBB [189], thus the arachnoid barrier establishes a BCSFB between the dura mater and the subarachnoid space.

The arachnoid mater is divided into two layers—the arachnoid barrier cell layer and the arachnoid trabeculae, extending into the subarachnoid space. The arachnoid barrier cell layer is characterized by closely joined cells connected by an extensive and continuous system of TJs [190] [191,192]. On the other hand, the arachnoid trabecular cells are characterized by the presence of sporadic fibroblasts [1,10,184]. The molecular composition of arachnoid barrier TJs is not yet well explored and may depend on the origin of these cells from the neural crest or mesoderm, which may lead to the development of epithelial or mesothelial junctional complexes, respectively. Recent studies have shown positive immunostaining for claudin-11 in arachnoid barrier cells in the brains of both humans and rodents [8,189], suggesting that the arachnoid BCSFB establishes TJs similar to those at the ChP BCSFB. Arachnoid barrier cells also express AQP1, the water channels expressed by ChP BCSFB epithelial cells. In addition, existence of AJs between arachnoid barrier fibroblasts has been supported by detection of junctional localization of E-cadherin between the arachnoid cells of the brain in mice [55,193,194]. Presence of junctional proteins in the arachnoidal cells has been further supported as culture of human arachnoid granulations, which are described as projections of the arachnoid membrane into the dural sinuses, have demonstrated the presence of cytoskeletal and junctional proteins [194]. In addition, gap junctions and desmosomes ensure arachnoid barrier function. The presence of connexin 43 composed gap junctions provides electrical and metabolic coupling of arachnoid barrier cells relevant for the regulation of CSF passage across this barrier [194,195]. Desmosomes between arachnoid barrier epithelial or mesothelial cells establish mechanical stability of this barrier by anchoring these junctional structures to the intermediate filament network of the cells [194,196,197,198]. A potential functional impact on the different developmental origin of arachnoid barrier cells from the neural crest or mesoderm and thus the possible maturation of epithelial versus mesothelial junctional complexes has not yet been investigated. Thus, a detailed analysis of region-specific differences in the molecular composition of arachnoid barrier junctional complexes and their role in regulating barrier formation remains to be performed.

Below the arachnoid barrier there is a subarachnoid space filled with cerebrospinal fluid. Towards the brain and spinal cord tissue the subarachnoid space is lined by the cells forming the pia mater. The pia mater is composed of evenly flattened cells that line all the surface of the brain and spinal cord. The cells of the pia mater lack TJs and do not form a barrier for solutes. At the same time, they are joined by desmosomes and gap junctions [199]. In accordance to the arachnoid barrier, the precise junctional compositions and the regional differences in their composition due to the different developmental origin of the pia mater cells remains to be investigated.

The pia mater is separated from the brain and spinal cord parenchyma by glia limitans formed by the parenchymal basement membrane and astrocyte endfeet [1,184]. In the healthy CNS there is no TJs between astrocyte endfeet, which are rather linked by gap junctions and not well-defined intercellular junctions [190]. Bradbury and colleagues speculated already in 1975 that this suggests that the astrocytic glia limitans establishes a border between the CSF and the brain parenchyma [190]. This notion is further supported by the observation that during neuroinflammation, when BBB integrity is impaired, reactive astrocytes can form TJs possibly prohibiting the parenchymal entry of humoral and cellular factors from the blood stream [200,201]. TJs in reactive astrocytes are composed of JAM-A, claudin-1 and claudin-4. While both claudins have been shown to play a vital role in glial scar formation, JAM-A rather seems to regulate the migration of immune cells across the glia limitans [201,202].

## 6. Visualization of the Junctional Complexes of the Brain Barriers

Recent advances in high end in vivo imaging techniques including two-photon intravital microscopy (2P-IVM) have allowed for unprecedented observations of immune cell entry into the CNS during immune surveillance and neuroinflammation. We have proposed that the brain barriers play a vital role in separating compartments in the CNS that provide different access to the immune cells. Advanced imaging has also allowed for precise localization of cellular junctional molecules [203,204] in vitro and in vivo. Exploring the visualization of molecularly distinct junctional complexes in epithelial, endothelial, mesothelial and glial barriers thus provides a novel option for visualization of these barriers by fluorescent imaging using immunofluorescence staining or transgenic reporter mouse models. Furthermore, visualization of junctional complexes in these barriers allows for the study of cellular pathways of immune cell trafficking across these barriers using in vivo live cell imaging [205]. Here, we provide a brief overview of the main tools allowing for reliable visualization of CNS barrier components and thus CNS compartments by confocal and 2P-IVM imaging in healthy and neuroinflammatory conditions. In that context, several published studies have employed endothelial and epithelial junctional reporter mice models to study immune cell trafficking into the CNS.

There are few major approaches allowing for visualization of junctional complexes in brain barriers. First, transgenic mouse models, where a fluorescent reporter is expressed under the promoter of the junctional protein expressed in the respective cell, can be employed. In this case, the soluble fluorescent protein will be visible in the whole volume of the cells expressing the corresponding junctional protein. For example, claudin-5, which has been identified as a major constituent of the TJs of CNS endothelial cells, was used to create the Tg (Cldn5-GFP) Cbet/U reporter mouse line [44], where green fluorescence protein (GFP) is expressed under the control of the claudin-5 promoter. The strong GFP signal in the CNS endothelial cells makes this mouse model very useful for direct visualization of CNS blood vessels by 2P-IVM (Figure 5).

Alternatively, knock-in mouse models can be employed where the fluorescent protein is fused to the respective junctional molecule. This approach enables visualization of the subcellular localization of the respective junctional molecule in the cells via the fluorescent tag. For example, the VE-cadherin-GFP knock-in mouse [206], expressing a C-terminal GFP fusion protein of VE-cadherin in the endogenous VE-cadherin locus, enables live imaging of vascular AJs. This model provides a GFP signal, the strength of which is dependent on endogenous VE-cadherin expression. It is suitable for 2P-IVM of vascular AJs in blood vessels of the brain and spinal cord of healthy mice, but also in mice suffering from experimental autoimmune encephalomyelitis (aEAE), an animal model for multiple sclerosis (Figure 6). In the dura mater, in addition to the blood vessels lined by VE-cadherin expressing vascular endothelial cells, the lymphatic vessels composed of lymphatic endothelial cells, expressing lower levels of VE-cadherin, can be visualized using the VE-cadherin-GFP knock-in reporter mouse model. VE-cadherin-GFP+ blood vessels can be distinguished from VE-cadherin GFP+ lymphatic vessels by the absence of signal of a fluorescent tracer such as orange TRITC Dextran (500 kDa) injected into the blood stream prior to imaging (Figure 6, red arrows). The VE-cadherin-GFP knock-in mouse is thus highly suitable for visualization of the endothelial AJs in the brain and spinal cord along the entire vascular tree. Additionally, it allows for imaging and distinguishing the dural blood and lymphatic vessels on the surface of the brain.

Similarly, the E-Cadherin-CFP knock-in mouse expressing the fluorescent protein monomer Cyan (mCFP) fused to the C-terminus of E-cadherin enables visualization of the AJs of epithelial cells [207]. This mouse model may be potentially suitable to visualize AJs between neural crest derived arachnoid barrier cells in the frontal brain, as a recent report demonstrated specific expression of E-cadherin in AJs between arachnoid barrier cells [55]. Once in vivo imaging techniques for the ChP are established it may also be used for visualization of AJs complexes of the ChP BCSFB in vivo.

In vivo imaging of BBB TJ complexes in the mouse has already been successfully reported [205]. To this end a reporter mouse expressing claudin-5 fused to eGFP under the control of the Tie-2 promoter was established [205]. This mouse model has allowed for visualization of claudin-5 in BBB TJs in vivo and direct observation of the cellular pathway of T-cell migration across the BBB owing to the strong expression of eGFP-CLN-5. Potential shortcomings of these models are that claudin-5-GFP is expressed under the control of the Tie-2 promoter, activity of which is regulated by inflammatory mediators. In addition, Tie-2 driven expression is not restricted to endothelium but allows for expression of the transgene in myeloid cells. This may impact on the molecular composition of the BBB TJ and thus its integrity and may be significantly different from physiological conditions.

Another shortcoming of the fusion protein models is the potential impact of the relatively big fluorescent proteins on the appropriate localization of junctional molecules and their interaction with intracellular scaffolding proteins. This issue can be addressed by employing self-complementing split fluorescent proteins [208,209], where only a small part of the florescent protein is used to tag the complex of interest, for example the GFP_11_ fragment of only 16 amino acids formed of a single β-strand in case of the GFP_1–10D7/11M3 OPT_ split protein [210]. The GFP1-10 fragment overexpressed in the corresponding cell remains non-fluorescent until complementation with the GFP_11_ fused to the target protein.

Additional fluorescent reporter mouse models can be obtained with the versatile aid of the Cre/loxP system. For example, by crossing the Ai14 line [211], where the fluorescent protein is expressed upon deletion of the loxP-flanked stop cassette in the Rosa26 locus, with lines expressing Cre under a brain barrier specific promoter [212], will allow to produce reporter mouse models with the respective fluorescent reporter.

Visualization of the BCSFB can be achieved by employing the FOXJ1-GFP reporter mouse or FOXJ1 Cre models [213], specific for the ciliated epithelial cells, or the transthyretin reporter [214], specific for the ChP epithelial cells.

To distinguish capillary and post-capillary segments of the CNS vascular tree, differential visualization of smooth muscle cells and pericytes can be used. For example, the expression of Cre and eGFP under the promoter of smooth muscle myosin heavy chain was developed to visualize the vascular and non-vascular smooth muscle cells [215]. Visualization of pericytes can be achieved with the Pdgfrb-Cre mouse models [216] or the NG2 reporter mouse lines [212]. In the latter cases, one has however to take into account that expression of the reporters may not be entirely exclusive to the brain barrier cells. NG2 is additionally expressed in oligodendrocyte precursor and smooth muscle cells, and Pdgfrb is also found to be expressed in oligodendrocytes, fibroblasts and smooth muscle cells.

Finally, in peripheral vascular beds visualization of junctional components has been achieved by intravenous injection of fluorescently tagged non-function blocking antibodies targeting, e.g., PECAM-1 to label endothelial cell junctions in vivo [217]. This approach may fail to detect junctional components in the complex and tight BBB TJs in the absence of neuroinflammation, as the antibodies may fail to reach their target. However, during neuroinflammation, when BBB TJ integrity is impaired this may provide a rapid approach for in vivo imaging of BBB junctions.

At the surface of the brain or spinal cord assignment of a vascular structure to a specific layer within the meninges or a compartment in the CNS can be facilitated by making use of the second harmonic generation (SHG) signal [218] obtained from collagen type I secreted in large amounts mainly by the fibroblasts in the dura mater. In 2P-IVM, using a short-pulse laser with an excitation wavelength in the range of 800–1300 nm, the excitation light can be separated from the emitted SHG signal, the wavelength of which is half of the excitation wavelength. SHG allows for localization of CNS compartments rich in collagens, especially the dura mater and, to a certain degree, the arachnoid barrier and pial layers (Figure 6). Furthermore, in some cases these compartments can be distinguished by the collagen fibers morphology [219] and can serve as a reliable landmark in the context of visualization of brain barriers by 2P-IVM.

Various small molecule tracers, such as TRITC and FITC Dextran, nanoparticles can be employed to visualize the luminal compartment of blood or lymphatic vessels. In neuroinflammation, imaging junctional complexes combined with the injection of vascular fluorescent tracers may allow to image BBB junctional leakiness. By injecting tracers into the ventricles or into the cisterna magna, distribution of the fluorescent tracers has been successfully applied to visualize paravascular CSF drainage pathways along subarachnoid vessels [220] allowing to determine CSF flow and the distribution of humoral factors in the CNS. Combining these methodologies with reporter mice for the brain barriers will allow advances in our understanding of CNS immune surveillance.

## 7. Conclusions

Despite decades of research, there are still a lot of unresolved questions concerning the precise molecular composition and interaction of TJs and AJs of the brain barriers, which have hampered the development of therapeutic strategies targeting the brain barriers in neuroinflammatory disorders. Regarding the BBB, it has been recently confirmed by several reports that claudin-1 and claudin-3 are absent from BBB endothelial cells and thus, claudin-5 is the only critical TJ claudin proven to date to contribute for the mouse BBB function [102]. However, when BBB endothelial cells lack caudin-5, TJs remain morphologically intact [92], which indicates that another claudin must maintain TJ morphology as visualized by transmission electron microscopy. Since we have recently reported that also claudin-12 is not required for proper BBB TJ function [106], the protein that maintains morphologically intact BBB TJs in the absence of claudin-5 remains to be identified. This approach is currently hampered by the lack of suitable antibodies detecting claudins with high and reliably specificity.

Structural and cellular junctions of meningeal barriers are less studied as compared to the BBB and the BCSFB. Meningeal barriers from the front and posterior region of the nervous system have been described to develop from different origins and therefore will have a different molecular make-up of their respective junctional complexes which may lead to regional differences in meningeal barrier functions.

A first set of available tools for in vivo visualization of the junctional components of the different brain barriers allows to study the dynamic interactions of immune cells with the brain barriers during health and neuroinflammation. Identifying the junctional components of the endothelial, epithelial, meningeal and glial brain barriers will set the stage for developing further mouse models, which will allow for in vivo imaging of the brain barriers. With these tools, one can thus explore their function in maintaining CNS immune surveillance and the impact of brain barrier disruption in neuroinflammation in vivo.

## Figures and Tables

**Figure 1 ijms-20-05372-f001:**
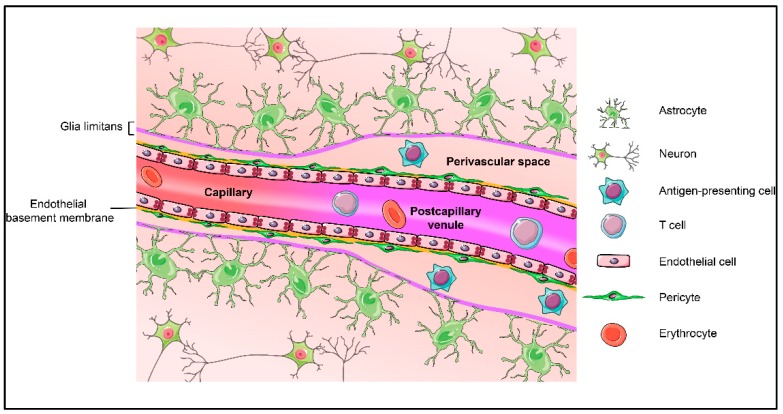
Schematic representation of the components of the neurovascular unit at the level of brain capillaries and post-capillary venules. Drawings of the individual cell types were adapted from Servier Medical Art (http://smart.servier.com/), licensed under a Creative Common Attribution 3.0 Generic License.

**Figure 2 ijms-20-05372-f002:**
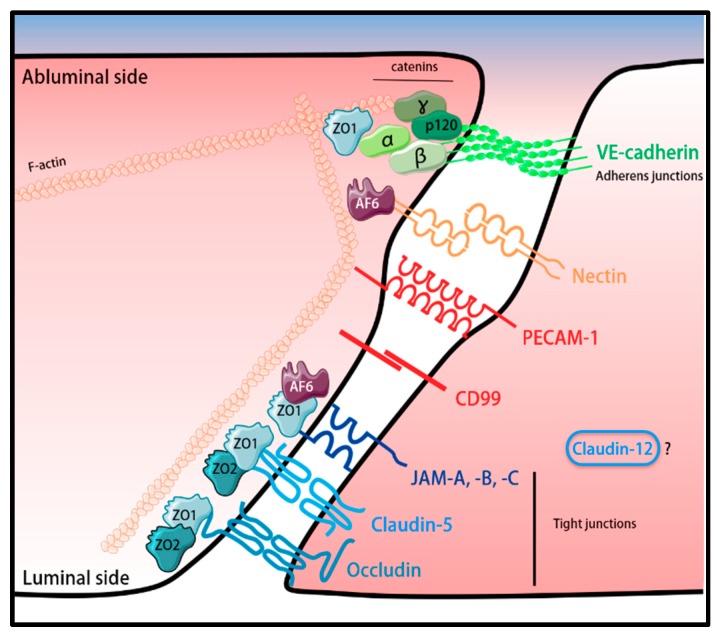
Schematic representation of the junctional complexes of the BBB. The proteins that compose the tight and adherens junctions are connected to the cytoskeleton via intracellular scaffolding proteins, ZO-1, ZO-2 and AF-6. Despite being expressed at low levels by the BBB endothelial cells [106], the subcellular location and function of claudin-12 remains to be defined. The forms of the individual proteins were adapted from Servier Medical Art (http://smart.servier.com/), licensed under a Creative Common Attribution 3.0 Generic License.

**Figure 3 ijms-20-05372-f003:**
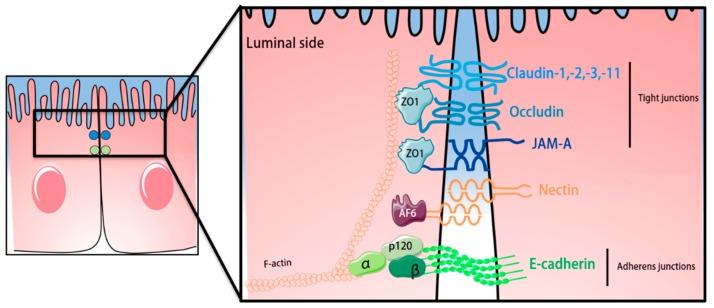
Schematic representation of the junctional complexes of the BCSFB at the choroid plexus. In similarity to the BBB, the proteins that compose the tight and adherens junctions of the choroid plexus BCSFB are connected to the cytoskeleton via intracellular scaffolding proteins and are localized in the apical part of the choroid plexus epithelial cells. The shapes of the proteins were adapted from Servier Medical Art (http://smart.servier.com/), licensed under a Creative Common Attribution 3.0 Generic License.

**Figure 4 ijms-20-05372-f004:**
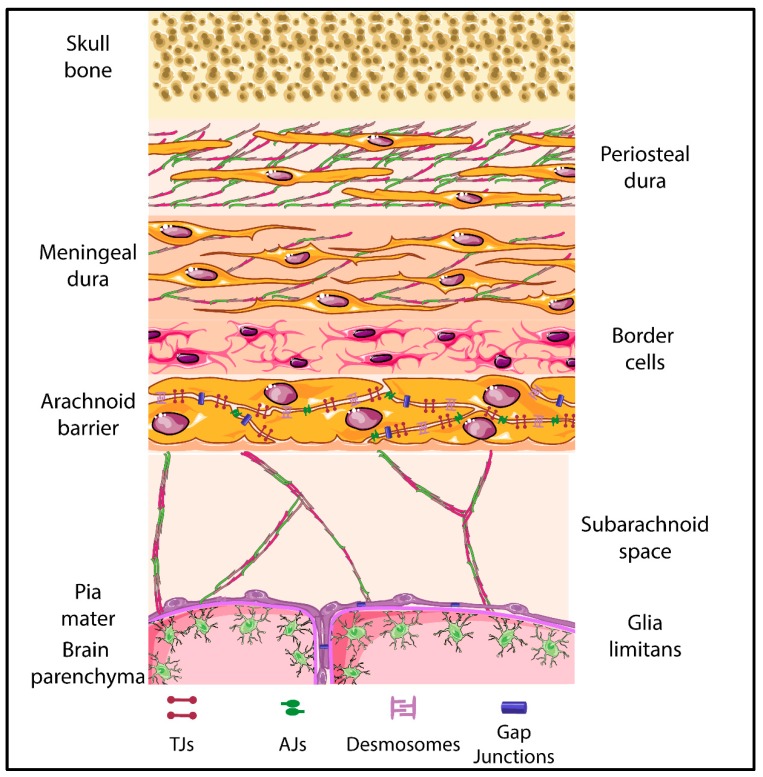
Schematic representation of the meningeal layers. The different layers and cellular composition of the meningeal layers are displayed. Tight junctions are highlighted between the arachnoid barrier cells as parallel lines, with adherens junctions, desmosomes and gap junctions also being represented. The shapes of the cell types were adapted from Servier Medical Art (http://smart.servier.com/), licensed under a Creative Common Attribution 3.0 Generic License.

**Figure 5 ijms-20-05372-f005:**
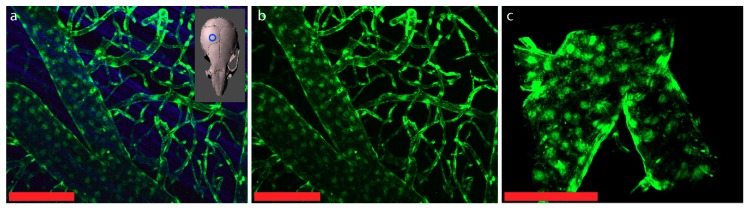
Representative images of the cranial and spinal cord window 2P-IVM imaging of the claudin-5-GFP reporter mouse. (**a**) The cranial window was placed over the right hemisphere of the mouse brain as depicted on the insert. Second harmonic generation in blue derives from collagen fibers in the dura mater. The strong GFP signal visible throughout the brain endothelial cells allows for imaging of the vessels along the entire vascular tree. (**b**) Cranial window region from (a) after removal of the dura mater. (**c**) Cervical spinal cord window. High magnification image showing the dorsal vein and branching veins. Scale bars = 100µm.

**Figure 6 ijms-20-05372-f006:**
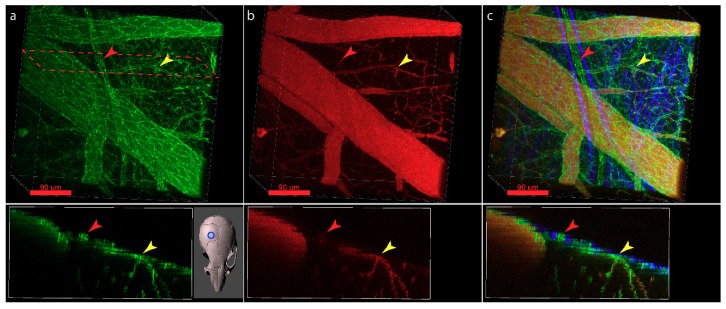
Visualization of CNS and meningeal endothelial AJs using the VE-Cadherin-GFP reporter mouse. (**a**) Cranial window allowing to visualize meningeal, subarachnoid, subpial and cortical vascular VE-cadherin-GFP+ AJs. (**b**) Cranial window preparation from (a) highlighting blood vessels after i.v. injection of TRITC Dextran. (**c**) Overlay of a and b and the second harmonic generation of the collagen fibers in the dura allow to distinguish VE-cadherin-GFP+ TRITC+ blood vessels and VE-cadherin-GFP+ TRITC^neg^ lymphatic vessels in the dura mater. Examples of blood and lymphatic vessels of similar caliber size are highlighted with a yellow and red arrowhead, respectively. The cranial window was placed over the right hemisphere of the mouse brain as depicted in the insert below (a). Top row—3D stack, bottom—XZ maximum intensity projection of 20 µm along Y at the cross section highlighted at the top. Scale bars = 90µm.

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
