# Peer review of "Structure and Junctional Complexes of Endothelial, Epithelial and Glial Brain Barriers"

_ijms, 2019, doi:10.3390/ijms20215372_

Round 1

Reviewer 1 Report

The review manuscript by Dias etal provides an excellent overview of the glial and endothelial barriers in the brain. The schematics are very well generated and offer a visual insight into the brain barrier mechanisms.

Few issues need attention.

1) The abstract as written is not compelling and sometimes confusing as to what the readers will expect from the review. This section needs to be revised. For example, the last sentence is a run-on sentence and does not indicate whether this issue is discussed in the manuscript (reporter mouse models is not the focus of the review).

In the abstract The authors should indicate – that they provide a brief overview of the main tools allowing for reliable visualization of CNS barrier components and thus CNS compartments by confocal etc..

2) Page 2, Lines 4-7. Please re-write the last sentence.

3) Page 2, topic 2: Please start with explaining NVU and then move to the BBB and its cellular components.

4) Rewrite sentence in lines 18-19 page 2.

5) page 3 line 9- please check for reference formats, here ref 30 is indicated twice.

6) Page 3 line 26- properties od? Please spell check

7) Page 3 line 31- pericytes and endothelial cell secrete what? Please indicate here.

8) Page 7 line 25- expand VE here.

9) Page 9 line 32- Please rewrite the beginning of this paragraph. If this is true for all mammals then state –In mammals the dura mater etc.. if specific to humans, then state in humans….

Author Response

Point -by -point reply to Reviewer #1:

The review manuscript by Dias et al provides an excellent overview of the glial and endothelial barriers in the brain. The schematics are very well generated and offer a visual insight into the brain barrier mechanisms.

We thank the Reviewer for the positive and productive feed-back.

Few issues need attention.

1) The abstract as written is not compelling and sometimes confusing as to what the readers will expect from the review. This section needs to be revised. For example, the last sentence is a run-on sentence and does not indicate whether this issue is discussed in the manuscript (reporter mouse models is not the focus of the review).In the abstract the authors should indicate – that they provide a brief overview of the main tools allowing for reliable visualization of CNS barrier components and thus CNS compartments by confocal etc..

The abstract has been revised.

2) Page 2, Lines 4-7. Please re-write the last sentence.

We have corrected the sentence accordingly.

3) Page 2, topic 2: Please start with explaining NVU and then move to the BBB and its cellular components.

We understand the suggestion of the reviewer and agree that starting by clarifying the concept of NVU would be an alternative option. However, the main focus of the review is on the cells forming the brain barriers and thus remain in this scheme, we believe it is better  to first address the characteristics of the BBB endothelial cells proper, which are forming the barrier, and only then mention that the interaction of these endothelial cells with cellular and acellular components of the NVU especially as we also refer to the glia limitans etablished by astrocytes as an additional barrier later in the review.

4) Rewrite sentence in lines 18-19 page 2.

We have corrected the sentence accordingly.

5) page 3 line 9- please check for reference formats, here ref 30 is indicated twice.

We have removed the reference number 30, it was indeed repeated.

6) Page 3 line 26- properties od? Please spell check

We have corrected the typing error.

7) Page 3 line 31- pericytes and endothelial cell secrete what? Please indicate here.

Pericytes and endothelial cells are described to secrete extracellular matrix proteins that comprise the endothelial basement membrane.

8) Page 7 line 25- expand VE here.

We have indicated that VE stands for Vascular Endothelial, and it is also found corrected in the abbreviations.

9) Page 9 line 32- Please rewrite the beginning of this paragraph. If this is true for all mammals then state –In mammals the dura mater etc.. if specific to humans, then state in humans….

We have altered it and mention that this detailed analysis has only been performed in humans.

Reviewer 2 Report

This is a well written review about a topic that has received too little attention in the literature, namely, the brain barriers that have been established to maintain brain homeostasis and function. The authors point to several unresolved questions concerning the precise molecular composition and interaction of tight junctions and adherens junctions of the brain barriers, which has hampered the development of therapeutic strategies targeting the brain barriers in neuroinflammatory disorders. The authors provide a detailed review of the cells and molecules involved in the formation of these barriers.  Several minor issues need to be addressed as listed below.

Page 2, line 5: change ‘fibroblasts have been described establish functionally similar barriers’ to ‘fibroblasts have been described to establish functionally similar barriers’

P2, l14: do not use an abbreviation before explaining it in the text: in the section heading change NVU to Neurovascular Unit. Likewise, use Neurovascular Unit (NVU) in the figure legend of figure 1.

P2, l18: change ‘features allowing to restricts the free passage of ions’ to ‘features to restrict the free passage of ions’

P2, l35: please explain, what is Mfsd2a?

P3, l4: change ‘GLUT-1’ to ‘The glucose transporter GLUT-1’

P3, l26: change ‘od’ to’of’

P3, l28: change ‘cells’ to ‘cell’

P3, l46: what are you referring to with ‘clearance of axonal material’?

P5, l28: change ‘these dramatic failure’ to ‘this dramatic failure’ or ‘these dramatic failures’

P6, l4: change ‘endothelial is’ to ‘endothelial cells is’

P7, l12: change ‘which is are essential’ to ‘which is essential’

P7, l29: change ‘that are connect’ to ‘that are connected’

P9, l17: define SLC transporters

P13, l23: change ‘modes’ to ‘model’

Author Response

Point by point reply to Reviewer 2

This is a well written review about a topic that has received too little attention in the literature, namely, the brain barriers that have been established to maintain brain homeostasis and function. The authors point to several unresolved questions concerning the precise molecular composition and interaction of tight junctions and adherens junctions of the brain barriers, which has hampered the development of therapeutic strategies targeting the brain barriers in neuroinflammatory disorders. The authors provide a detailed review of the cells and molecules involved in the formation of these barriers.  Several minor issues need to be addressed as listed below.

We thank the Reviewer for the positive and productive feed-back.

Page 2, line 5: change ‘fibroblasts have been described establish functionally similar barriers’ to ‘fibroblasts have been described to establish functionally similar barriers’

We have corrected the sentence accordingly.

P2, l14: do not use an abbreviation before explaining it in the text: in the section heading change NVU to Neurovascular Unit. Likewise, use Neurovascular Unit (NVU) in the figure legend of figure 1.

We have corrected the abbreviations throughout the manuscript and updated the abbreviations list.

P2, l18: change ‘features allowing to restricts the free passage of ions’ to ‘features to restrict the free passage of ions’

We have corrected the sentence accordingly.

P2, l35: please explain, what is Mfsd2a?

We have highlighted in the text what is Mfsd2a.

P3, l4: change ‘GLUT-1’ to ‘The glucose transporter GLUT-1’

We have added the requested information about GLUT-1.

P3, l26: change ‘od’ to’of’

We have corrected the typing error.

P3, l28: change ‘cells’ to ‘cell’

We have corrected the typing error.

P3, l46: what are you referring to with ‘clearance of axonal material’?

We have changed the word “material” to “debris” to clarify this statement.

P5, l28: change ‘these dramatic failure’ to ‘this dramatic failure’ or ‘these dramatic failures’

We have corrected the typing error.

P6, l4: change ‘endothelial is’ to ‘endothelial cells is’

We have corrected the typing error.

P7, l12: change ‘which is are essential’ to ‘which is essential’

We have corrected the typing error.

P7, l29: change ‘that are connect’ to ‘that are connected’

We have corrected the typing error.

P9, l17: define SLC transporters

We have added the definition of SLC transporters.

P13, l23: change ‘modes’ to ‘model’

We have corrected the typing error.

Reviewer 3 Report

The manuscript entitled “Structure and Junctional Complexes of Endothelial, Epithelial and Glial Brain Barriers” is a detailed and up to date review of the molecular make-up of the blood brain barriers. The authors make important points regarding the deficiencies of our knowledge about the tissue and subcellular localisation of specific claudin proteins which is largely caused by the small size of and high homology between the members of this protein family. The authors discuss the reliability of every method used so far for this purpose and thus the reliability of available claudin protein (and also mRNA) data. As the authors point out, transgenic mice expressing fluorescent fusion proteins is the way to address this problem in intravital microscopy, of course with the restrictions discussed in the manuscript well kept in mind.

One small remark on the topic of fluorescent fusion claudins is that the compact size of claudin and resulting complications of getting a functional fluorescent claudin warrants research exploring the use of smaller fluorescent tags and split fluorescent proteins such as the GFP1-10/11 system. Although in the latter the steric hindrance caused by the fluorescent protein tag is not removed but placed from claudin oligomerization to label assembly.

The manuscript is well written and suitable for publishing as is.

There are a handful of typing or grammar errors that do not affect the quality of the manuscript, but could be corrected by a pair of fresh eyes and careful proofreading. If there are other issues that warrant a minor revision please consider this as well. I will mention a few points to look at besides typing errors: The first word of Conclusions : “Despite” has to be followed by a noun, not a verb, thus something like “Despite decades of research,” would be the grammatically correct beginning for the sentence. Check the list of abbreviations, for example Cdh1 is listed, but does not appear in the manuscript. Check the use of comprise, comprise of, constitute.

Author Response

The manuscript entitled “Structure and Junctional Complexes of Endothelial, Epithelial and Glial Brain Barriers” is a detailed and up to date review of the molecular make-up of the blood brain barriers. The authors make important points regarding the deficiencies of our knowledge about the tissue and subcellular localisation of specific claudin proteins which is largely caused by the small size of and high homology between the members of this protein family. The authors discuss the reliability of every method used so far for this purpose and thus the reliability of available claudin protein (and also mRNA) data. As the authors point out, transgenic mice expressing fluorescent fusion proteins is the way to address this problem in intravital microscopy, of course with the restrictions discussed in the manuscript well kept in mind.

One small remark on the topic of fluorescent fusion claudins is that the compact size of claudin and resulting complications of getting a functional fluorescent claudin warrants research exploring the use of smaller fluorescent tags and split fluorescent proteins such as the GFP1-10/11 system. Although in the latter the steric hindrance caused by the fluorescent protein tag is not removed but placed from claudin oligomerization to label assembly.

We agree with the reviewer and have included a brief description and corresponding references about the split-fluorescent protein method.

The manuscript is well written and suitable for publishing as is.

There are a handful of typing or grammar errors that do not affect the quality of the manuscript but could be corrected by a pair of fresh eyes and careful proofreading. If there are other issues that warrant a minor revision please consider this as well. I will mention a few points to look at besides typing errors: The first word of Conclusions: “Despite” has to be followed by a noun, not a verb, thus something like “Despite decades of research,” would be the grammatically correct beginning for the sentence. Check the list of abbreviations, for example Cdh1 is listed, but does not appear in the manuscript. Check the use of comprise, comprise of, constitute.

We thank the reviewer for the very positive feedback. We have addressed the grammar errors throughout the manuscript, and we corrected the abbreviations and updated the abbreviations list.

Reviewer 4 Report

The review article by Dias and colleagues regards the fundamental aspects of tight junctions in different brain barriers. In general, the article was a very good read and it will have a great value for both experts and novices in the field. I have the following comments:

1. As also indicated by the choice of themes in the Conclusion, there are sections that are more interesting than others. For example, the section regarding claudins is fantastic and very nice too read with a great flow, whereas sections like that on adherens junctions are too enumerative to get a good flow in the text. I understand that the authors want to describe a lot of information, but the article will benefit from re-thinking some sections and decide whether they are crucial for the article or not. In the case of the adherens junction section, a simple shortening would probably do the work.

2. How do the authors choose what is state of the art? I am aware of the expertise of the authors' group, which will inevitably yield a lot of background knowledge to communicate. However, I find no indications that any form of systematic approaches were used to ensure that no relevant literature was left out.

3. The authors provide a very nice description of the blood-brain barrier structure. However, there is no mentioning of the endothelial glycocalyx, which is a shame. There is even recent work done using 2PM to study the glycocalyx as a diffusion barrier (Kutuzov et al. (2018), PNAS, (Martin Lauritzen group)). Could the authors please expand the article with a section on the glycocalyx?

4. The section on P- and E-face tight junctions could be shortened to half its original length without loss of quality.

5. All the references in the section on basement membranes are rather old. Of course, the Sorokin reference is an evergreen, but there are also newer review articles that ought to be cited. For example, Thomsen et al. (2017), J Cereb Blood Flow Metab (Torben Moos group), is rather extensive.

6. I think that the part on the pericyte's role in a low level of vesicular transport is a bit too short for its importance at the BBB. The authors could add Villasenor et al. (2016), Sci Rep (Roche), which illustrates this in a nice way.

7. In relation to the section about the pia mater and the subarachnoid space, I would like to hear the authors' comments to the structural findings in Pizzo et al. (2018), J Physiology (Robert Thorne group).

8. The section on 2PM is surprisingly short (with only few references) given its description as a main theme in the Introduction. I would suggest to change the title to encompass the study of the BBB in general and include a lot more references.

With reference to the ongoing discussion on reviewers demanding citations to their own work, I just want to state that I am not a co-author on any of the articles suggested above.

Author Response

The review article by Dias and colleagues regards the fundamental aspects of tight junctions in different brain barriers. In general, the article was a very good read and it will have a great value for both experts and novices in the field. I have the following comments:

As also indicated by the choice of themes in the Conclusion, there are sections that are more interesting than others. For example, the section regarding claudins is fantastic and very nice too read with a great flow, whereas sections like that on adherens junctions are too enumerative to get a good flow in the text. I understand that the authors want to describe a lot of information, but the article will benefit from re-thinking some sections and decide whether they are crucial for the article or not. In the case of the adherens junction section, a simple shortening would probably do the work.

We thank the reviewer for the very positive feedback. We have kept all the sections that we believe that contribute significantly to the logical flow of the manuscript, but we have shortened some sections.

How do the authors choose what is state of the art? I am aware of the expertise of the authors' group, which will inevitably yield a lot of background knowledge to communicate. However, I find no indications that any form of systematic approaches were used to ensure that no relevant literature was left out.

We understand the criticism of the Reviewer and agree that the overall scheme of the Reivew does not allow to include all relevant references, when solely considering the expression of claudins at the BBB. We have thus removed the term state of the art to avoid misconceptions.

The authors provide a very nice description of the blood-brain barrier structure. However, there is no mentioning of the endothelial glycocalyx, which is a shame. There is even recent work done using 2PM to study the glycocalyx as a diffusion barrier (Kutuzov et al. (2018), PNAS, (Martin Lauritzen group)). Could the authors please expand the article with a section on the glycocalyx?

We agree with the reviewer that mentioning the glycocalyx is of the utmost importance. Therefore, we have added a paragraph about the importance of the glycocalyx in the subsection referring to the BBB endothelial cells.

The section on P- and E-face tight junctions could be shortened to half its original length without loss of quality.

We have shortened the information we provide about P- and E-face associated tight junctions and added instead additional references.

All the references in the section on basement membranes are rather old. Of course, the Sorokin reference is an evergreen, but there are also newer review articles that ought to be cited. For example, Thomsen et al. (2017), J Cereb Blood Flow Metab (Torben Moos group), is rather extensive.

We agree with the reviewer and have added more recent references, including however again the most recent work from the Sorokin group who is leading in the field.  

I think that the part on the pericyte's role in a low level of vesicular transport is a bit too short for its importance at the BBB. The authors could add Villasenor et al. (2016), Sci Rep (Roche), which illustrates this in a nice way.

We agree with the reviewer and have elaborated more on the potential role of pericytes including the discussion about their potential role in regulating blood flow. We have also included additional including the suggested references.

In relation to the section about the pia mater and the subarachnoid space, I would like to hear the authors' comments to the structural findings in Pizzo et al. (2018), J Physiology (Robert Thorne group).

We agree with the reviewer and have briefly mentioned the work done by Pizzo et al in the section about meningeal brain barriers.

The section on 2PM is surprisingly short (with only few references) given its description as a main theme in the Introduction. I would suggest changing the title to encompass the study of the BBB in general and include a lot more references.

Please note that we have specifically included this paragraph to hilight the limited models available at this time to image the brain barriers and their junctional complexes in their entire dynamics. Thus we would prefer to keep the respective headline. We have however extended this paragraph and have also better specified the role of this paragraph in the context of this review in the abstract. We hope this is acceptable to this reviewer.  

Reviewer 5 Report

RE: IJMS-629145

The manuscript titled “Structure and Junctional Complexes of Endothelial, Epithelial and Glial Brain Barriers” by Dias et al represents a critical evaluation of the current knowledge on structure and function of the barriers in the CNS. Specifically, they identify a unique opportunity of the utilization of the state of the art techniques to study localization and functional role of different participants in each of the brain barriers both in health and disease conditions. It is a very nicely written review. Such overviews are important in disseminating current findings, and studies that offer the potential to extend the current knowledge and point the readers into the currently unknown areas to be explored.

Though, authors should add some information/discussion on pericytes. They mentioned that both pericytes and smooth muscle cells (SMC) express αSMA. However, not all populations of pericytes express αSMA, whereas neighboring cells such as SMCs do express αSMA. Also, pericytes from different branching order in BBB have different αSMA patterns. For more information refer to Armulik et al, Dev Cell 2011, Hill et al. JNIP 2014, Hill et al, Neuron 2015, Vanlandewijck M, et al. Nature 2018

Author Response

Reviewer #5:

The manuscript titled “Structure and Junctional Complexes of Endothelial, Epithelial and Glial Brain Barriers” by Dias et al represents a critical evaluation of the current knowledge on structure and function of the barriers in the CNS. Specifically, they identify a unique opportunity of the utilization of the state-of-the-art techniques to study localization and functional role of different participants in each of the brain barriers both in health and disease conditions. It is a very nicely written review. Such overviews are important in disseminating current findings, and studies that offer the potential to extend the current knowledge and point the readers into the currently unknown areas to be explored.

Though, authors should add some information/discussion on pericytes. They mentioned that both pericytes and smooth muscle cells (SMC) express αSMA. However, not all populations of pericytes express αSMA, whereas neighboring cells such as SMCs do express αSMA. Also, pericytes from different branching order in BBB have different αSMA patterns. For more information refer to Armulik et al, Dev Cell 2011, Hill et al. JNIP 2014, Hill et al, Neuron 2015, Vanlandewijck M, et al. Nature 2018

We thank the reviewer for the very positive feedback and suggestions. We have extended the section on mural cells and included the suggested references.